# Protective Effect of Oxygen and Isoflurane in Rodent Model of Intestinal Ischemia-Reperfusion Injury

**DOI:** 10.3390/ijms24032587

**Published:** 2023-01-30

**Authors:** Mathias Clarysse, Alison Accarie, Ricard Farré, Emilio Canovai, Diethard Monbaliu, Jan Gunst, Gert De Hertogh, Tim Vanuytsel, Jacques Pirenne, Laurens J. Ceulemans

**Affiliations:** 1Department of Abdominal Transplant Surgery & Transplant Coordination, University Hospitals Leuven, 3000 Leuven, Belgium; 2Abdominal Transplant Laboratory, Department of Microbiology, Immunology and Transplantation, KU Leuven, 3000 Leuven, Belgium; 3Leuven Intestinal Failure and Transplantation Center (LIFT), University Hospitals Leuven, 3000 Leuven, Belgium; 4Translational Research Center for Gastrointestinal Disorders (TARGID), Department of Chronic Diseases and Metabolism (CHROMETA), KU Leuven, 3000 Leuven, Belgium; 5Department of Intensive Care Medicine, University Hospitals Leuven, 3000 Leuven, Belgium; 6Laboratory of Intensive Care Medicine, Department of Cellular and Molecular Medicine, KU Leuven, 3000 Leuven, Belgium; 7Department of Pathology, Laboratory of Translational Cell & Tissue Research, University Hospitals Leuven, KU Leuven, 3000 Leuven, Belgium; 8Department of Gastroenterology and Hepatology, University Hospitals Leuven, 3000 Leuven, Belgium; 9Department of Thoracic Surgery, University Hospitals Leuven, 3000 Leuven, Belgium; 10Laboratory of Respiratory Diseases and Thoracic Surgery (BREATHE), Department of Chronic Diseases and Metabolism (CHROMETA), KU Leuven, 3000 Leuven, Belgium

**Keywords:** anesthesia, intestinal ischemia, ischemia-reperfusion injury, outcome, oxygen, survival

## Abstract

Animal research in intestinal ischemia-reperfusion injury (IRI) is mainly performed in rodent models. Previously, intraperitoneal (I.P.) injections with ketamine–xylazine mixtures were used. Nowadays, volatile anesthetics (isoflurane) are more common. However, the impact of the anesthetic method on intestinal IRI has not been investigated. We aim to analyze the different anesthetic methods and their influence on the extent of intestinal IRI in a rat model. Male Sprague–Dawley rats were used to investigate the effect of I.P. anesthesia on 60 min of intestinal ischemia and 60 min of reperfusion in comparison to hyperoxygenation (100% O_2_) and volatile isoflurane anesthesia. In comparison to I.P. anesthesia with room air (21% O_2_), supplying 100% O_2_ improved 7-day survival by cardiovascular stabilization, reducing lactic acidosis and preventing vascular leakage. However, this had no effect on the intestinal epithelial damage, permeability, and inflammatory response observed after intestinal IRI. In contrast to I.P. + 100% O_2_, isoflurane anesthesia reduced intestinal IRI by preventing ongoing low-flow reperfusion hypotension, limiting intestinal epithelial damage and permeability, and by having anti-inflammatory effects. When translating the aforementioned results of this study to clinical situations, such as intestinal ischemia or transplantation, the potential protective effects of hyperoxygenation and volatile anesthetics require further research.

## 1. Introduction

Intestinal ischemia-reperfusion injury (IRI) is a frequent clinical entity, typically observed in patients suffering from intestinal ischemia but also as an inevitable part of intestinal transplantation (ITx) [1,2].

Ischemia causes the depletion of adenosine triphosphate (ATP) and the accumulation of radical oxygen species (ROS) precursors [3]. When blood flow is restored, additional mucosal injury occurs because reperfusion exacerbates the ischemic insult through an inflammatory process known as IRI [4]. Reperfusion causes massive ROS formation, which activates molecular (chemokines/cytokines/coagulation cascade) and cellular (neutrophils/macrophages/platelets) components of innate immunity, leading to local and systemic inflammation and cell death [5,6,7]. As a consequence, IRI can disrupt the mucosal barrier, causing bacterial (endotoxin) translocation and sepsis [1,3,4,5,7,8,9]. Intestinal IRI is also inevitable in ITx, where it stimulates the alloimmune response, which negatively impacts the graft outcome [10].

Research to unravel the pathophysiologic mechanisms and potential interventions in intestinal IRI is mainly performed in rodent models [11,12]. Previously, intraperitoneal injections with ketamine–xylazine mixtures were used as an anesthetic method [13,14,15]. However, more recently, volatile anesthetics came into practice because it is easier to apply and reproduce and it improves survival rates [14]. In rodent experiments, isoflurane is mainly used as a volatile anesthetic, and it is administered in conjunction with pure oxygen (100%) [14,16]. Ketamine anesthesia in intestinal IRI has shown protective effects on epithelial injury, as well as anti-inflammatory and antithrombotic properties, compared to intraperitoneal pentobarbital anesthesia [15,17,18,19]. Hyperoxia, by pure oxygen administration before, during, and after intestinal IRI experiments with intraperitoneal pentobarbital anesthesia, has been shown to reduce small bowel injury, accelerate enterocyte turnover, and improve intestinal rehabilitation without affecting survival in rodents [20,21]. Isoflurane is mainly known as a potent anti-inflammatory agent and has shown protective effects in a post-conditioning setting in intestinal IRI [14]. These properties have been tested in liver and kidney transplantation in large multicenter clinical trials. The results of the volatile conditioning of the transplant procedure were non-inferior in regard to delayed graft function compared to intravenous anesthetics [22,23]. However, there was a decreased incidence of acute rejection at 2 years post-transplant in the volatile anesthesia group of kidney transplants [23].

However, to our knowledge, the impact of these anesthetic methods in se has not been investigated in intestinal IRI. Therefore, we aimed to analyze the different potential modulators, which might influence the outcome of intestinal IRI in a rodent model. As volatile anesthetics are provided with oxygen supplementation, we split the experimental setup to investigate, on one hand, the effect of oxygen supplementation, followed, on the other hand, by the additional potential protective effect exerted by isoflurane.

## 2. Results

### 2.1. Oxygen Supplementation

#### 2.1.1. Oxygen Supplementation Improved Cardiovascular Stability and Prevented Acidosis in Intestinal IRI

In a rodent model of 60 min of atraumatic clamping of the superior mesenteric artery, applying 100% O_2_ increased the oxygen saturation and had a stabilizing effect on the heart rate during the experiment (Figure 1A,C). Partial oxygen pressure (pO_2_) measurements at sacrifice showed hypoxia in non-oxygenated animals, while 100% O_2_ administration led to hyperoxia: sham ischemia with intraperitoneal anesthesia (I.P.) (sham I.P. + 21% O_2_) 52.38 ± 15.95 mmHg and I.P. + 21% O_2_ 73.48 ± 20.84 mmHg vs. I.P.+100% O_2_: 268 ± 22.31 mmHg (both *p* < 0.001) (Figure 1B). When looking at systolic blood pressure, a significant drop was seen at the moment of reperfusion, 60 min after the onset of ischemia, despite the use of oxygen supplementation (Figure 1D). Significant difference in pH was seen between sham and I.P. + 21% O_2_ at the moment of sacrifice after 60 min of reperfusion: 7.31 ± 0.02 vs. 7.16 ± 0.10 (*p* = 0.02), whilst this was not different from I.P. + 100%: 7.19 ± 0.08 (*p* = 0.10 and *p* > 0.99, respectively) (Figure 1E). By analyzing lactate, a significant difference was noted between non-oxygenated and oxygenated animals: 4.07 ± 1.10 mmol/L vs. 1.23 ± 0.16 mmol/L (*p* < 0.001) (Figure 1F).

#### 2.1.2. Oxygen Supplementation Did Not Protect against Intestinal Damage

IRI had a severe impact on the intestinal wall integrity, as shown by its histological evaluation via the Park–Chiu score and the villus height (Figure 2A,B). The Park–Chiu score after IRI in the I.P. + 21% O_2_ group was 5.83 ± 0.41, which was not altered following oxygen supplementation (I.P. + 100% O_2_ 6.00 ± 0.00) (*p* > 0.99). This was confirmed by the villus height, which measured 95.2 ± 19.3 µm with I.P. + 21% O_2_ versus 105.1 ± 9.2 µm for I.P. + 100% O_2_ (*p* > 0.99). When looking at the intestinal epithelial permeability to ions, by measuring the transepithelial electrical resistance (TEER) in an Ussing chamber setup, no difference was noted: 7.5 ± 2.8 Ω *cm^2^ in I.P. + 21% O_2_ versus 8.6 ± 1.8 Ω *cm^2^ in the I.P. + 100% O_2_ group (*p* = 0.99; Figure 2C).

#### 2.1.3. Oxygen Supplementation Preserved Vascular Permeability and Prevented Endotoxin Translocation Following IRI

When assessing the endothelial glycocalyx, a significant increase was measured in plasmatic syndecan-1 levels after IRI without oxygen supplementation: 129.6 ± 33.48 ng/mL vs. 93.26 ± 11.04 ng/mL (*p* = 0.02). Plasmatic heparan sulfate levels showed a significant increase after IRI in I.P. + 21% O_2_ in contrast to sham: 233.5 ± 109.1 ng/mL vs. 46.63 ± 54.35 ng/mL (*p* = 0.05) (Figure 3A,B). This impaired endothelial glycocalyx structure was also demonstrated by (i) an increased vascular leakage, as shown by hemoconcentration: 17.8 ± 1.3 vs. 13.4 ± 1.0 g/dL (*p* = 0.02), (ii) increased reperfusion edema (wet/dry ratio): 6.15 ± 1.18 vs. 4.41 ± 0.397 (*p* = 0.001), and (iii) increased plasmatic endotoxin levels: 83.13 ± 32.89 vs. 20.14 ± 8.94 mU/mL (*p* < 0.001) (Figure 3C,E).

#### 2.1.4. Oxygen Supplementation Did Not Prevent the Inflammatory Response Provoked by IRI

Following intestinal IRI with I.P. anesthesia with 21% of oxygen, plasmatic levels of interleukin (IL)-6 significantly increased in contrast to sham treatment: 1540.0 ± 856.3 vs. 334.0 ± 165.6 pg/mL (*p* = 0.001). Oxygen supplementation had no impact on IL-6 levels: 1022 ± 568.3 pg/mL (*p* = 0.34 and *p* = 0.11, respectively) (Figure 4A). This inflammatory reaction was also seen at the gene transcription level in tissues. Tissue IL-1β levels were significantly altered after IRI, irrespective of oxygen supplementation: 1.036 ± 0.2985 (sham I.P. + 21% O_2_) vs. 6.837 ± 3.572 (I.P.+21% O_2_, *p* = 0.002) and vs. 8.497 ± 3.269 (I.P. + 100% O_2_, *p* < 0.001) (Figure 4B). Similar results were observed for the anti-inflammatory cytokine IL-10: 1.019 ± 0.2035 (sham I.P. + 21% O_2_) vs. 14.58 ± 14.03 (I.P. + 21% O_2_, *p* = 0.23) and vs. 23.77 ± 16.38 (I.P. + 100% O_2_, *p* = 0.01) (Figure 4C), and TNF-α: 0.9035 ± 0.4588 (sham I.P. + 21% O_2_) vs. 4.211 ± 1.141 (I.P. + 21% O_2_, *p* = 0.006) and vs. 5.288 ± 1.330 (I.P. + 100% O_2_, *p* < 0.001) (Figure 4D).

### 2.2. Isoflurane Anesthesia

#### 2.2.1. Isoflurane Anesthesia Prevented Reperfusion Hypotension in Intestinal IRI and Acidosis

Irrespective of whether anesthesia was intraperitoneal (I.P.+100% O_2_) or volatile (isoflurane), no differences were seen in oxygen saturation or heart rate if 100% of oxygen was applied during the experiment (Figure 5A,C). pO_2_ measurements showed more pronounced hyperoxia in the isoflurane ischemic rats (isoflurane + 100% O_2_): sham ischemia with isoflurane (sham isoflurane + 100% O_2_) 249.0 ± 70.57 mmHg and I.P. + 100% O_2_ 268.0 ± 22.31 mmHg vs. isoflurane + 100% O_2_: 354.7 ± 63.84 mmHg (*p* = 0.004 and *p* = 0.02, respectively) (Figure 5B). However, the systolic blood pressure decreased significantly in the I.P. group but not in the isoflurane group (Figure 5D). A significant difference in pH was seen between sham isoflurane and I.P. + 100% O_2_: 7.322 ± 0.03 vs. 7.193 ± 0.08 (*p* = 0.04), whilst this was not different from isoflurane + 100% O_2_: 7.238 ± 0.04 (*p* = 0.17 and *p* > 0.99, respectively) (Figure 5E). Lactate levels were similar across groups (Figure 5F).

#### 2.2.2. Isoflurane Protected against Histopathological Damage and Preserved Intestinal Permeability, Provoked by IRI

IRI clearly impacted the intestinal tissue; however, this was less severe when isoflurane was used in contrast to I.P. anesthesia: Park/Chiu score 3 ± 1.549 (gas + 100% O_2_) vs. 6 ± 0 (I.P. + 100% O_2_) (*p* = 0.47) (Figure 6A,D–F). This was confirmed by reduced loss of villus height with isoflurane anesthesia: 191.3 ± 65.72 vs. 105.1 ± 9.227 µm (*p* = 0.77) (Figure 6B). The measurement of TEER revealed a decreased intestinal epithelial permeability to ions as well: 20.30 ± 7.098 Ω *cm^2^ in gas + 100% O_2_ vs. 8.6 ± 1.8 Ω *cm^2^ in I.P. + 100% O_2_ (*p* < 0.001; Figure 6C).

#### 2.2.3. Isoflurane Did Not Additionally Protect Vascular Permeability or Reduce Endotoxin Translocation in the Case of Oxygen Supplementation

No differences were noted in endothelial glycocalyx structure, hemoglobin concentration, and reperfusion edema when isoflurane was used (Figure 7A–D). Endotoxin translocation was also not altered: 8.277 ± 0.4748 mU/mL (gas + 100% O_2_) vs. 20.14 ± 8.935 mU/mL (*p* = 0.67) (Figure 7E).

#### 2.2.4. Isoflurane Impeded the Inflammatory Response Following IRI

Plasmatic levels of IL-6 were significantly reduced with isoflurane anesthesia: 136.7 ± 52.52 pg/mL (gas + 100% O_2_) vs. 1022.0 ± 568.3 pg/mL (I.P. + 100% O_2_) (*p* = 0.02) (Figure 8A). This was also reflected in significantly reduced IL-1β tissue mRNA levels: 2.865 ± 2.021 (gas + 100% O_2_) vs. 8.497 ± 3.269 (I.P. + 100% O_2_, *p* = 0.003) (Figure 8B). This effect was also present but less pronounced for the anti-inflammatory cytokine IL-10 and TNF-α, mainly due to high variations (Figure 8C–D).

### 2.3. Oxygen Supplementation and Isoflurane Improved Survival Following IRI

When 7-day survival was assessed in the rodent model of 60 min of intestinal ischemia, no survival was seen with I.P. anesthesia with 21% oxygen (air) in contrast to 70% survival with oxygen supplementation and even 90% when isoflurane was used instead (*p* < 0.0001) (Figure 9).

## 3. Discussion

In this experimental rodent study on intestinal IRI, we studied the effect of oxygen supplementation and isoflurane anesthesia (volatile) compared to I.P. ketamine–xylazine anesthesia.

Oxygen supplementation clearly ameliorated the detrimental effects of intestinal IRI with ketamine–xylazine anesthesia, with significantly improved 7-day survival. The protective effects of hyperoxia (100% O_2_) in intestinal IRI experiments have been shown before. The main effects were attributed to decreased enterocyte apoptosis and decreased neutrophil recruitment by the downregulation of E-selectin production [20,21]. In this study, we show survival benefits after intestinal IRI, which were mainly mediated by improved vascular permeability, as the endothelial glycocalyx (eGC) was better preserved. Endothelial glycocalyx destruction is now considered a cornerstone for the detrimental effects attributed to IRI [9,24]. As fewer constituents of the eGC were detected in the plasma, a better preservation of the eGC can be assumed. This was confirmed by the fact that the vascular permeability/leakage was improved, as shown by reduced hemoglobin concentration, decreased reperfusion edema, and decreased endotoxin translocation. These measurements were not explained by an altered intestinal epithelial permeability (TEER) nor by histopathological alterations. Hemodynamically, oxygen supplementation resulted in a less depressed heart rate, which presumably led to a better cardiac output throughout the whole experiment. This could explain the less pronounced lactic acidosis, which is normally seen after intestinal IRI. However, it could not counteract the hypotensive episode observed at the moment of reperfusion, which could lead to an ongoing, low-flow ischemic episode in this experimental setting. There was no protective effect of oxygen supplementation on inflammation, which might be mediated through increased ROS formation [25]. However, in IRI studies, hyperoxygenation has been interpreted with caution due to the paradoxical idea of increasing ROS formation. On the other hand, in recent IRI studies, it has been shown that hyperoxygenation might actually protect against IRI by a net favorable effect on plasma oxidative status and hence reduce ROS formation. This effect seems to be mediated by the activation of pro-inflammatory cascades by hyperoxia, which includes interference with neutrophils adhesion and free radical production [20,21].

Volatile anesthetics with isoflurane provided an additional survival benefit over injection anesthesia with ketamine–xylazine and hyperoxia. The protective effects of isoflurane on intestinal IRI have only been shown in a treatment setting so far, where mainly protective effects on epithelial injury were seen [26]. However, isoflurane is known as a potent anti-inflammatory agent, such as in renal IRI [27]. This study confirms the potent anti-inflammatory properties of isoflurane as both systemic and local inflammatory cytokines were significantly reduced. This anti-inflammatory effect has been described to be mediated by the activation of the peroxisome-proliferator-activated receptor gamma/nuclear factor-kappa B pathway (PPARγ/NF-κB). PPARγ has been reported to ameliorate LPS-induced inflammation through the TLR4 signaling pathway [28]. Such anti-inflammatory response to intestinal IRI has also been seen with opioids (remifentanil), as shown by the study of Cho et al. [29]. Secondly, we confirmed the protective effect of isoflurane on intestinal epithelial damage, as was previously shown in a treatment setting [26]. The destruction of the intestinal epithelial cells seems to be prevented by increased transforming growth factor-beta1 (TGF-β1) production induced by isoflurane exposure [26]. This epithelial protective effect has also been seen with vitamins (folic acid, alphatocopherol), anti-apoptotic drugs (ruboxustaurin and caveolin-1), opioids (remifentanil), and stem cell therapy [11,12,29,30,31]. In our study, the improved preservation of the intestinal epithelial lining was also confirmed by reduced intestinal epithelial permeability (TEER). As to the vascular permeability, no difference was seen. Hemodynamically, there was the known vasodilatory effect of isoflurane, immediately from the start of the experiment. However, with volatile anesthesia, there was no hypotensive episode at the moment of reperfusion. As such, there was no persisting low-flow ischemia at the moment of reperfusion in the volatile anesthesia group, in contrast to I.P. injection with ketamine–xylazine. Volatile anesthesia, such as isoflurane, has also been shown to reduce ROS formation in intestinal IRI by reducing malondialdehyde (MDA) production [28]. ROS is also known to modulate the JAK/STAT pathway, and protective effects of isoflurane through activating this pathway have been shown in cardiac IRI [32,33].

The study inherently has limitations. First, although animals received I.P. fluid resuscitation, they were not mechanically ventilated and did not receive intravenous fluid resuscitation and vasopressive support to treat hypotension. The animals not receiving oxygen supplementation had mild to moderate hypoxemia. Hence, it is not clear whether the protective effects could be achieved by preventing hypoxemia rather than by achieving hyperoxemia. Likewise, it is not clear if more aggressive treatment of postreperfusion hypotension could have prevented the observed harm. Second, only isoflurane and oxygen administration were studied and no other common anesthetics. For example, pentobarbital has been a common anesthetic in rodent experiments as well. In a study by Kawai et al., it was shown that pentobarbital reaches a lower depth of anesthesia in comparison to ketamine–xylazine. On the other hand, ketamine/xylazine appears to work faster and lasts longer than pentobarbital [34]. More common clinical anesthetics, such as a mixture with nitrous oxide, which might have beneficial effects, are not tested in this study. Nitrous oxide could help in hemodynamic stabilization and has analgesic effects. In contrast, the use of nitrous oxide can have several adverse effects, such as pneumonia, atelectasis, increased skin infection, and sepsis [35,36]. Third, we did not study the impact of isoflurane vs. ketamine–xylazine in the absence of oxygen supplementation. Hence, we do not know whether the observed protection by isoflurane is dependent on oxygen supplementation. However, the study by Wilding et al. showed that using either 21% or 100% of oxygen as a carrier for isoflurane anesthesia revealed no major differences in main physiologic parameters. With 100% oxygen, hypercapnia leading to relative hypertension, decreased respiratory rates, and more pronounced respiratory acidosis was more commonly seen in comparison to 21% [16]. Further, structural changes were seen in the endothelial glycocalyx, in combination with secondary, indirect effects. However, these structural changes do not warrant mechanistic or pathophysiologic advantages per se. Lastly, the current findings are limited to male rats, as in these intestinal IRIs is more pronounced than in their female counterparts [37,38].

This study has unmasked the substantial influence of the anesthetic method in a rodent model of intestinal IRI. Volatile anesthetics resulted in fewer hemodynamical changes compared to ketamine–xylazine anesthesia. Future implications of this can lead to a better choice of anesthetic strategy in more complex rodent experimental models, such as transplant models, in which preserved hemodynamics is crucial for the outcome. When translating the aforementioned results of this study to clinical situations, such as intestinal ischemia, hyperoxygenation might lead to a significant survival benefit and volatile anesthetics might give an additional survival benefit. IRI is inherent to transplant surgery, and according to the results of this study, intestinal transplant recipients might benefit from hyperoxygenation and volatile anesthetics. This beneficial effect of volatile anesthetics over injection anesthetics has already been shown in liver and kidney transplant recipients but warrants further research [22,23].

## 4. Materials and Methods

### 4.1. Animal Model

Male Sprague–Dawley rats (Janvier Labs, Saint Berthevin Cedex, France), weighing 275–350 g, 6 weeks old, were housed in the KU Leuven animal facility under specific pathogen-free conditions, with 2–3 rats per cage. The rats were acclimatized for 5–7 days before any intervention. The animals were kept in 14/10 h light/dark cycles, controlled temperature, and they received rat chow and water ad libitum. The animals were not fasted before surgery. Institutional animal research ethical committee approval—following the EU directive for animal experiments—was granted under the number P122/2019. The reported animal study is in compliance with the ARRIVE guidelines 2.0 [39].

### 4.2. Anesthesia

Intraperitoneal (I.P.) injection: mixture of 57.14 mg/kg body weight (BW) of ketamine (100 mg/mL, Nimatek, Eurovet, The Netherlands) and 5.71 mg/kg BW of xylazine (Xyl-M 2%, Inovet, Belgium) [38].

Gas: isoflurane (1000 mg/g, Iso-Vet, Dechra, Belgium), induction with 5% and maintenance with 1.5–1.75% at 1 L/min, and 100% O_2_ by usage of an isoflurane vaporizer (Harvard Apparatus, Holliston, MA, USA).

### 4.3. Surgery

IRI was performed with 60 min of ischemia and 60 min of reperfusion. Intestinal IRI was induced, after median laparotomy of 4 cm on the linea alba, by isolated atraumatic clamping of the superior mesenteric artery. Ischemia was checked by pulselessness in the mesentery and discoloration/dysmotility of the bowel. The laparotomy wound was temporarily closed during the experiment. At reperfusion, 1 mL of warmed saline (37 °C) was administered intraperitoneally to compensate for fluid loss by evaporation. Reperfusion was checked by recovery of arterial pulsations in the mesentery, recoloration, and regain of motility. At the end of the experiment, all animals were sacrificed by exsanguination, followed by blood and intestinal tissue collection. The animals were under anesthesia until exsanguination. All experiments were performed by the same researcher (M.C.).

Rats were randomly divided into five groups (*n* = 6/group) (Table 1, Figure 10):

For survival analysis, 10 additional animals were included in each IRI group and observed daily for 7 days. The laparotomy wound was closed subcutaneously in 2 layers with Prolène 4-0 (Ethicon, Belgium), and 0.5 mL of ropivacaine (3.16 mg/kg BW, 2 mg/mL, Naropin, Aspen, Ireland) was administered in the wound edges for local analgesia.

In accordance with animal welfare, rats were monitored at least 3 times daily, and buprenorphine subcutaneous (0.016 mg/kg BW, 0.3 mg/mL, Vetergesic, Ceva, Belgium) was used for analgesia, once preoperatively and twice daily postoperatively, during the first 3 days.

At the end of the experiment, rats were anesthetized with pentobarbital before sacrifice (65 mg/kg BW, 200 mg/mL, Dolethal, Vetoquinol, Belgium).

### 4.4. Vital Signs

Vital parameters were measured, every 15 min, from 5 min before onset of ischemia until awakening/sacrifice/death. Rectal temperature was measured by usage of a clinical thermometer (SC19 flex rapid, SCALA, Frankfurt, Germany). Heart rate and saturation were measured on hind paws by usage of OxiPen (EnviteC, Germany). Tail cuff blood pressure measurements were performed by usage of the Coda non-invasive blood pressure system (Kent Scientific, Torrington, CT, USA). Ten blood pressure measurements were taken at each time point, and the median was used for analysis.

### 4.5. Blood and Tissue Sampling

Heparinized blood samples were collected after puncture of the aorta for blood gas analysis (0.4 mL) in 2 EDTA tubes. The tubes were spun at 3500 rpm for 10 min at 4 °C. Plasma was snap frozen in liquid nitrogen and stored at −80 °C. One ileal tissue sample of 5 cm was taken just proximally of the ileocaecal valve, kept in glucose buffer on ice, and mounted in the Ussing chambers. Ileal samples were collected proximally of the previous and preserved in 4% buffered formalin for histological evaluation and snap frozen, after feces removal, for molecular analysis.

### 4.6. Arterial Blood Gas Analysis

Hemoglobin levels (g/dL), hematocrit (%), pH, arterial oxygen partial pressure (pO_2_) (mmHg), arterial carbon dioxide partial pressure (pCO_2_) (mmHg), oxygen saturation (%), and plasmatic L-lactate release (mmol/L) were analyzed by blood gas analyzer (ABL-815, Radiometer, Copenhagen, Denmark).

### 4.7. Histological Evaluation

Full-thickness samples were formalin-fixed, paraffin-embedded, cut into 5 µm coupes, and stained with hematoxylin–eosin. The ischemic injury was scored in a blinded fashion by an experienced pathologist (G.D.H.) on four fields per section by usage of the Park–Chiu score [40,41].

Villus length—defined as the distance between the mouth of the crypts and the tip of the villi—was measured in 4 different fields per tissue section, and the average was calculated to avoid the potential impact of patchy necrosis.

### 4.8. Ussing Chamber Experiments

#### Electrophysiological Parameters

Full-thickness ileal tissue (mucosa, submucosa, muscular layer, and serosa) was mounted, in triplicate, in a standard vertical Ussing chamber (Mussler Scientific Instruments, Aachen, Germany) with an opening of 9.60 mm^2^ by a blinded, experienced researcher (A.A.). Each half chamber was filled with 3 mL Krebs solution with 10 mM mannitol at the mucosal side and 10 mM glucose at the serosal side. Both buffers were maintained at 37 °C and continuously oxygenated with 95%/5% O_2_/CO_2_ and stirred by gas flow in the chambers. In this setup, data sampling and pulse inductions are computer-controlled using Clamp software (Version 9.00, Mussler Scientific Instruments, Aachen, Germany). Transepithelial electrical resistance (TEER) was measured by averaging 90 min of measurement after initial 30 min of stabilization. All tissue was mounted within 10 min after the exsanguination of the animal.

In our particular setting of intestinal IRI, leading to a diminished/denudated mucosal surface area (as shown by Grootjans et al. [42]), TEER was corrected by multiplying TEER with its corresponding villus length divided by the average villus length of the sham group [13].

### 4.9. Endotoxin Levels

Quantification of plasma endotoxin levels was obtained by the colorimetric limulus amebocyte lysate test (LAL QCL-1000^TM^, Lonza, Belgium), according to the manufacturer’s instructions. Absorbance was measured spectrophotometrically by FLUOstar Omega (BMG Labtech, Offenburg, Germany) at 410 nm. Corrections were made by the subtraction of the absorbance of the sample without the addition of LAL.

### 4.10. Edema

Tissue water content (edema) was assessed by the ratio between the weight before and after drying. Snap-frozen, whole-thickness ileal tissue samples were weighed just before and immediately after drying them for 3 h at 80 °C in a drying oven with forced convection (VENTI-Line VL 115, VWR, Belgium). The results were expressed as a wet/dry ratio and a percentage of water in the tissues ((1-(Dry weight/Wet weight))*100).

### 4.11. ELISA

Plasma Syndecan-1 (E-EL-R0996, Elabscience, TX, USA) concentration was measured by ELISA according to the manufacturer’s instructions.

Plasma heparan sulfate (OKEH02552, Aviva Systems Biology, San Diego, CA, USA) concentration was measured by ELISA according to the manufacturer’s instructions.

Plasma IL-6 concentration was measured by enzyme-linked immunosorbent assay (ELISA) according to the manufacturer’s instructions (R6000B, Bio-Techne Ltd., Abingdon, UK).

### 4.12. Quantitative Reverse-Transcription Polymerase Chain Reaction (qRT-PCR)

The relative expression of pro-inflammatory cytokines (interleukin (IL)-1β, tumor necrosis factor (TNF)-α, and anti-inflammatory cytokine (IL-10)) were determined by qRT-PCR. Tissue was homogenized in TRIzol reagent (Life Technologies, Carlsbad, CA, USA), and total ribonucleic acid (total RNA) was extracted using the RNeasy isolation kit (Qiagen, Germantown, MD, USA), according to the manufacturer’s instructions. cDNA was synthesized from 200 ng total RNA using M-MLV transcriptase (Life Technologies, Carlsbad, CA, USA). Next, real-time PCR reaction was performed on a LightCycler 96W (Roche, Vilvoorde, Belgium) with Taqman Fast Universal PCR Master Mix and Taqman gene expression assays (Applied Biosystems, Life Technologies, Carlsbad, CA, USA) (IL-1β (Rn00580432_m1), TNF-α (Rn00562055), and IL-10 (Rn00563409)). A three-step amplification program was used: 95 °C for 10 min, followed by 45 cycles of amplification (95 °C for 10 s, 60 °C for 15 s, 72 °C for 10 s). Target messenger RNA (mRNA) expression for cytokines was quantified relative to the housekeeping gene GAPDH (Life Technologies, Carlsbad, CA, USA) and to the sham using the “-ΔΔCt method”.

### 4.13. Statistical Analysis

All data were expressed as mean ± standard deviation and represented in scattered plots. The line in the middle is plotted at the mean. The whiskers indicate the standard deviation. Data were checked for outliers by the ROUT method with Q = 1% and subjected to (log)normality testing (Shapiro–Wilk test). Comparisons between multiple groups were performed with one-way ANOVA and post-hoc Tukey’s test in the case of normal distribution or the Kruskal–Wallis and post-hoc Dunn tests for non-normal distribution. Survival analysis was performed by the Kaplan–Meier test (log-rank test). A *p*-value < 0.05 was considered statistically significant (GraphPad Prism version 9.4.0 for Windows, GraphPad Software, San Diego, CA, USA).

## 5. Conclusions

In experimental rodent models, different types of anesthesia are commonly used. In this study on intestinal IRI, it was shown that oxygen supplementation had a protective effect on the vascular permeability of the intestine. Isoflurane anesthesia attenuated the detrimental effects of intestinal IRI even more by reducing intestinal epithelial permeability and the inflammatory cascade compared to ketamine–xylazine anesthesia. The type of anesthesia used in these experimental models can influence the outcome parameters analyzed and should be taken into account. The potential clinical implications should be investigated in ischemic and transplant patients.

## Figures and Tables

**Figure 1 ijms-24-02587-f001:**
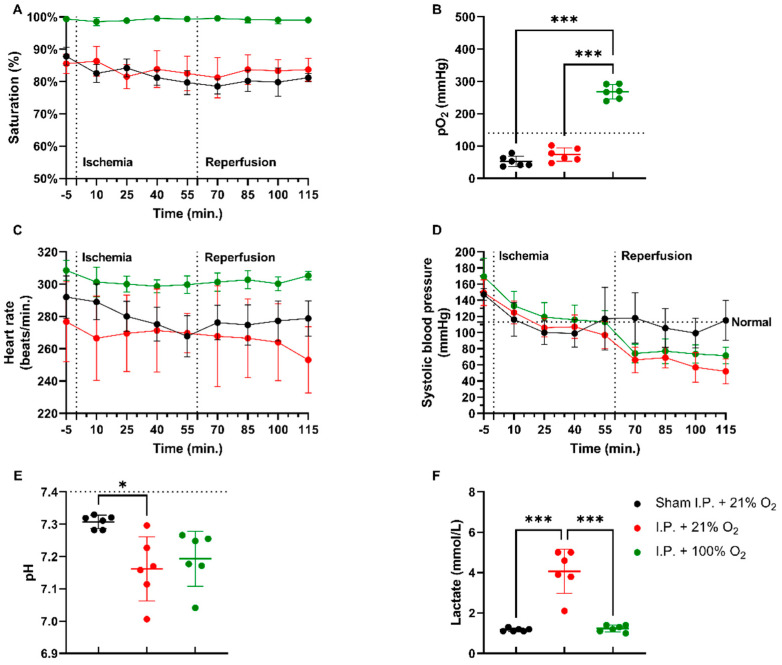
Hemodynamic changes were monitored throughout the 60 min of ischemia and 60 min of reperfusion. The impact of oxygen supplementation was assessed on saturation (**A**), partial oxygen pressure (pO_2_) (**B**), heart rate (**C**), and systolic blood pressure (**D**). After 60 min of reperfusion, arterial blood gas analysis revealed important pH (**E**) and lactate (**F**) changes. (*n* = 6/group). I.P.: intraperitoneal injection anesthesia; O_2_: oxygen. Normal values are shown by a horizontal dashed line in graphs (**C**,**D**). Onset of ischemia and reperfusion are shown by vertical dashed lines in graphs (**A**–**C**). * *p* < 0.05; *** *p* < 0.001.

**Figure 2 ijms-24-02587-f002:**
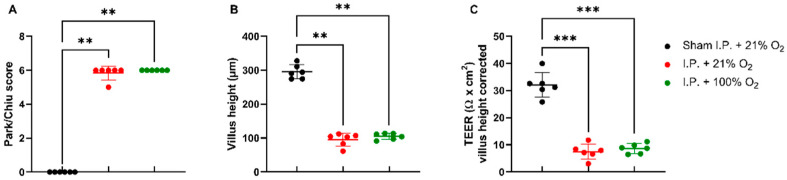
The impact of oxygen supplementation on intestinal epithelial injury was scored according to the Park–Chiu score (**A**) and villus height (**B**). Intestinal epithelial permeability was measured by TEER in an Ussing chamber setup, which was corrected for the villus height (**C**). (*n* = 6/group). I.P.: intraperitoneal injection anesthesia; O_2_: oxygen; TEER: transepithelial electrical resistance. ** *p* < 0.01; *** *p* < 0.001.

**Figure 3 ijms-24-02587-f003:**
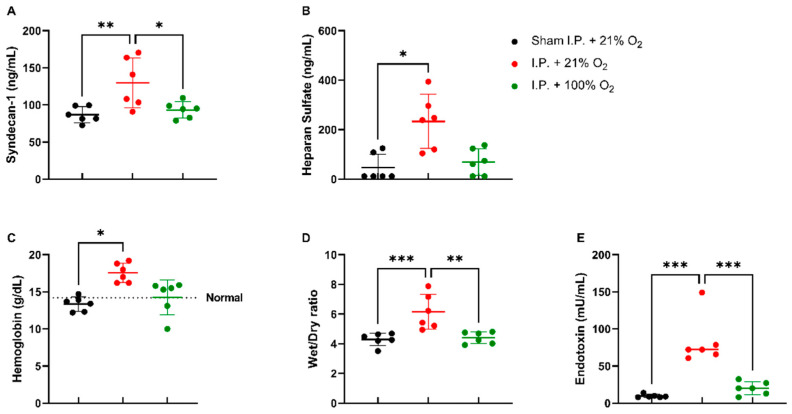
Vascular barrier changes due to oxygen supplementation were assessed by measuring plasmatic endothelial glycocalyx components: syndecan-1 (**A**) and heparan sulfate (**B**). Secondary effects in the form of hemoglobin concentration (**C**), reperfusion edema (wet/dry ratio, (**D**)), and bacterial translocation (plasmatic endotoxin levels, (**E**)) were measured (*n* = 6/group). I.P.: intraperitoneal injection anesthesia; O_2_: oxygen. Normal value is shown by a horizontal dashed line in graph (**C**). * *p* < 0.05; ** *p* < 0.01; *** *p* < 0.001.

**Figure 4 ijms-24-02587-f004:**
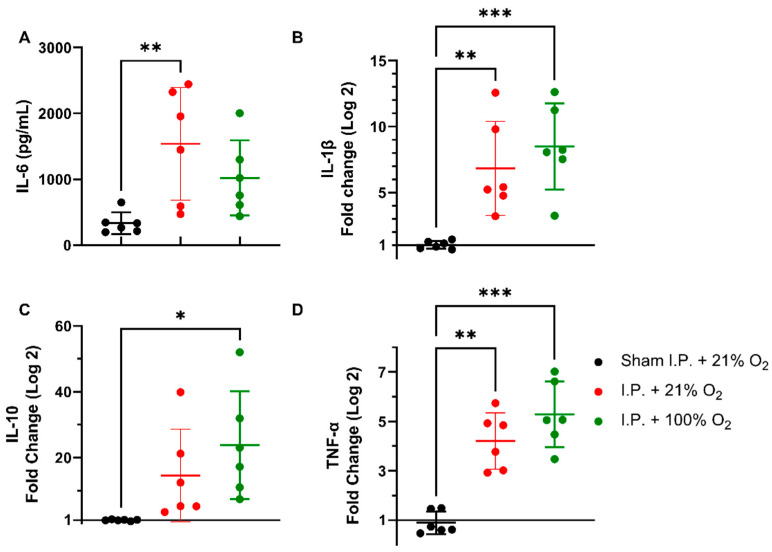
Inflammatory changes in systemic IL-6 (**A**) and intestinal IL-1β (**B**), IL-10 (**C**), and TNF-α (**D**) due to oxygen supplementation were assessed. (*n* = 6/group). IL: interleukin; I.P.: intraperitoneal injection anesthesia; O_2_: oxygen; TNF-α: tumor necrosis factor—alfa. * *p* < 0.05; ** *p* < 0.01; *** *p* < 0.001.

**Figure 5 ijms-24-02587-f005:**
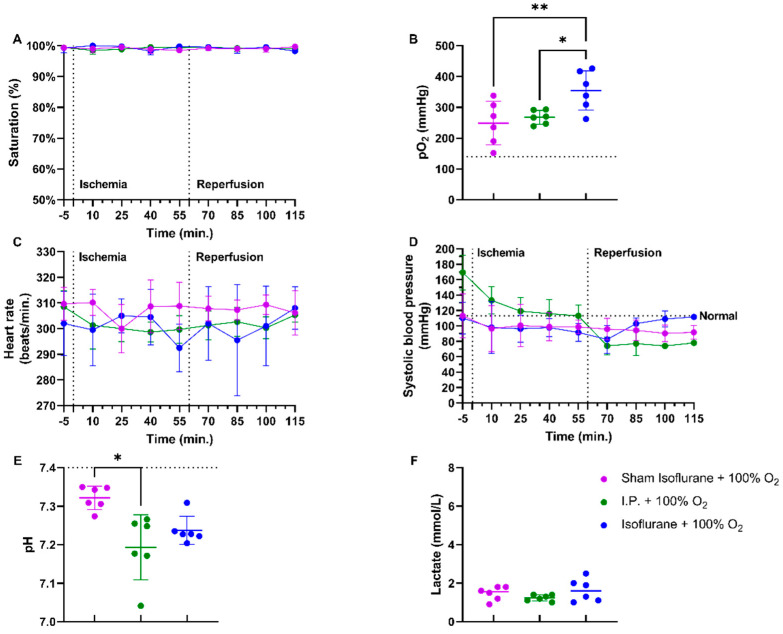
Impact of isoflurane on hemodynamic parameters during intestinal ischemia-reperfusion injury: saturation (**A**), partial oxygen pressure (pO_2_) (**B**), heart rate (**C**), and systolic blood pressure (**D**). After 60 min of reperfusion, pH (**E**) and lactate (**F**) were measured on arterial blood gas. (*n* = 6/group). I.P.: intraperitoneal injection anesthesia; O_2_: oxygen. Normal values are shown by a horizontal dashed line in graphs (**C**,**D**). Onset of ischemia and reperfusion are shown by vertical dashed lines in graphs (**A**–**C**). * *p* < 0.05; ** *p* < 0.01.

**Figure 6 ijms-24-02587-f006:**
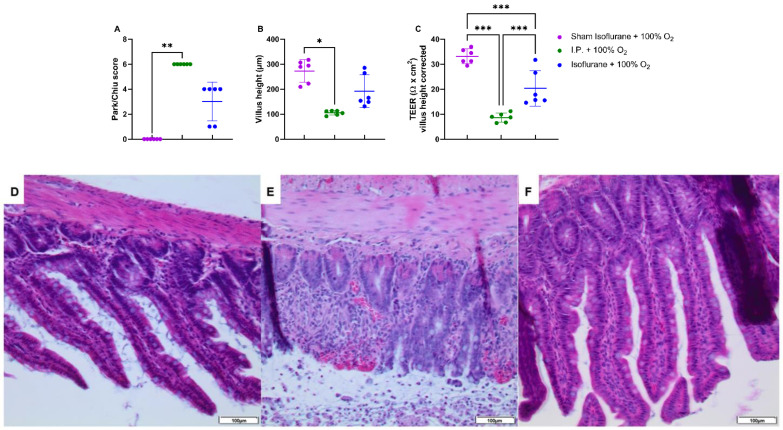
Intestinal epithelial injury was assessed by Park–Chiu score (**A**) and villus height (**B**). Histopathological pictures after hematoxylin–eosin staining are shown in (**D**–**F**): (**D**): Sham isoflurane + 100% O_2_; (**E**): I.P. + 100% O_2_; (**F**): isoflurane + 100% O_2_. Intestinal epithelial permeability was measured by TEER in an Ussing chamber setup, which was corrected for the villus height (**C**). (*n* = 6/group). I.P.: intraperitoneal injection anesthesia; O_2_: oxygen; TEER: transepithelial electrical resistance. * *p* < 0.05; ** *p* < 0.01; *** *p* < 0.001.

**Figure 7 ijms-24-02587-f007:**
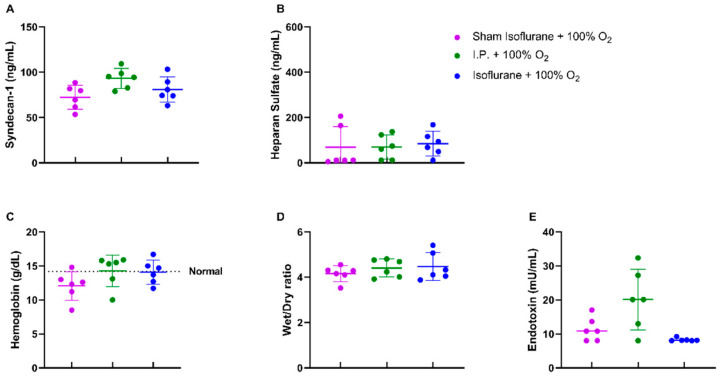
Isoflurane anesthesia did not have a protective effect on the endothelial glycocalyx either directly: plasmatic syndecan-1 (**A**) and heparan sulfate (**B**) or indirectly: hemoglobin concentration (**C**), reperfusion edema (wet/dry ratio, (**D**)), and bacterial translocation (plasmatic endotoxin levels, (**E**)). (*n* = 6/group). I.P.: intraperitoneal injection anesthesia; O_2_: oxygen. Normal value is shown by a horizontal dashed line in graph (**C**).

**Figure 8 ijms-24-02587-f008:**
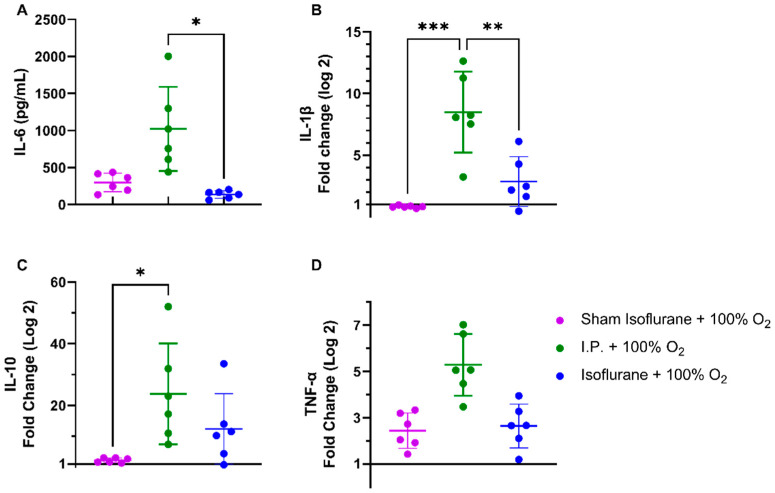
Isoflurane exerted anti-inflammatory changes on systemic IL-6 (**A**) and intestinal IL-1β (**B**), IL-10 (**C**), and TNF-α (**D**). (*n* = 6/group). IL: interleukin; I.P.: intraperitoneal injection anesthesia; O_2_: oxygen; TNF-α: tumor necrosis factor—alfa. * *p* < 0.05; ** *p* < 0.01; *** *p* < 0.001.

**Figure 9 ijms-24-02587-f009:**
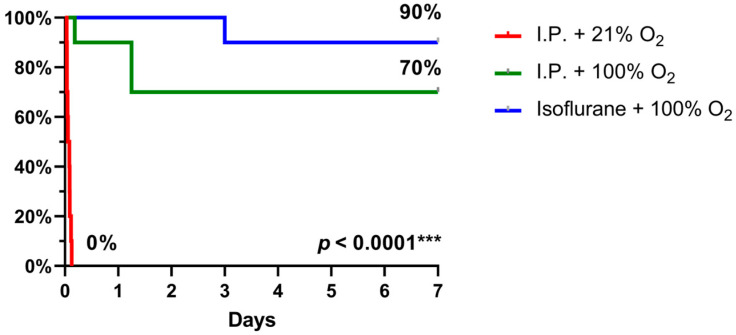
7-day survival, assessed by Kaplan–Meier analysis, revealed a significant benefit with oxygen supplementation and even more with isoflurane anesthesia. (*n* = 10/group). I.P.: intraperitoneal injection anesthesia; O_2_: oxygen.

**Figure 10 ijms-24-02587-f010:**
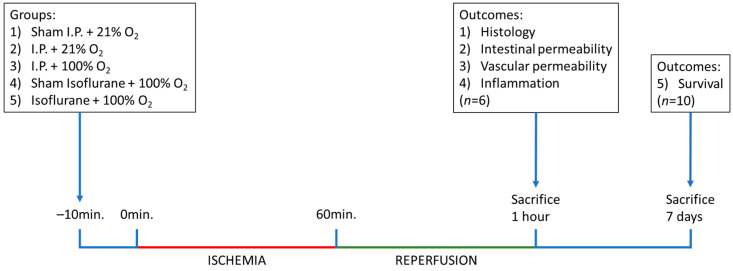
Timeline of the experiments, with sampling time points and explanation of the different groups. I.P.: intraperitoneal injection anesthesia; O_2_: oxygen.

**Table 1 ijms-24-02587-t001:** Experimental groups.

Group	Ischemia	Oxygen	Anesthesia
Sham I.P. + 21% O_2_	Sham	21%	Ketamine–Xylazine
I.P. + 21% O_2_	60 min	21%	Ketamine–Xylazine
I.P. + 100% O_2_	60 min	100%	Ketamine–Xylazine
Sham isoflurane + 100% O_2_	Sham	100%	Isoflurane
Isoflurane + 100% O_2_	60 min	100%	Isoflurane

## Data Availability

All data are available upon request from the corresponding author.

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
