# Peer review of "Protective Effect of Oxygen and Isoflurane in Rodent Model of Intestinal Ischemia-Reperfusion Injury"

_ijms, 2023, doi:10.3390/ijms24032587_

Round 1

Reviewer 1 Report

The manuscript submitted by Clarysse et al. (ijms-1903672) shows a study on the assessment of protective effect of oxygen and isoflurane in a model of intestinal ischemia-reperfusion injury. They proved that isoflurane anesthesia limits damage to the intestinal epithelium.

The study seems to be focused not only on cases of mesenteric ischemia-reperfusion, but also on the field of intestinal transplantation. In this regard, the proposed model could be made more complex by performing an intestinal anastomosis, making it more similar to the surgical technique of an intestinal transplant (taking into account that it is not possible to perform a vascular anastomosis of the mesenteric vessels in a rat model).

The manuscript is well written, but some important aspects, concerning methodological issues as well as the discussion of the results, need to be revised, which will undoubtedly improve the quality and soundness of the manuscript.

·      The introduction section should also include state-of-the-art details on the therapeutic management of IRI, either at the experimental or clinical level.

·      When describing the results, it would be easier to read if the values were spaced off the "±" sign.

·      Regarding survival studies, Figure 9 appears to show that none of the animals in the I.P.+21% O2 group survive longer than the immediate postoperative period (a few hours). The death of all the animals so soon after only 1 hour of ischemia and 1 hour of reperfusion seems to be an excessive damage. In previous experiences of our group, at 48 hours, mortality in untreated animals was 75%, even with longer ischemia times (2 hours). How do the authors explain such a high mortality with milder ischemia (the bibliographic reference is attached, although it is only available in Spanish) (http://www.oc.lm.ehu.es/laboratorio/Publicaciones/PDF%20articulos/1992%20CirEsp2.pdf).

·      Although the authors reported the weight of the animals, they also should indicate the age of the animals.

·      Why was this dose of ketamine chosen? Published studies to test the effect of these drugs in a model of myocardial ischemia reperfusion in rats (10.1177/0023677215597136) use doses of 75 mg/kg of ketamine (1.3 times more than that used in this work).

·      In those groups treated with oxygen, why did they use only oxygen and not a mixture with nitrous oxide as used in the clinical setting? Insofar as the analgesic and anesthetic effects of nitrous oxide make it a valuable complement (doi.org/10.1093/bjaceaccp/mkv019; doi.org/10.1016/j.bpa.2018.06.003)

·      The discussion should be rephrased. The authors should discuss their results with other published therapies in the treatment of mesenteric IRI, such as antioxidant agents (folinic acid, allopurinol, nitroindazole, curcumin...).En lo relativo a los estudios de supervivencia.

I strongly encourage the authors to consider my comments, made on a critical and constructive basis, as I consider that the manuscript has great potential for publication in IJMS.

Yours faithfully,

Reviewer 2 Report

The paper by Clarysse et al., entitled “Protective effect of oxygen and isoflurane in rodent model of intestinal ischemia-reperfusion injury” showed that anesthesia via isoflurane with hyperoxygenation protects intestine against damage, driven in IRI. Although authors performed comprehensive in vivo study (based on a very demanding surgical animal model), research lacks key fundamental points that are already highlighted by the authors in the discussion section of the paper. As the researchers had all the necessary control groups in IP experiments, the group isoflurane+21% O2 is lacking, which is one of the most important experimental group. Also the sex and age of the animals are important factor that should be considered. What is also the mechanism behind the protection (ROS formation, TLR signaling??)? Also the whole paper is lacking some novel citation in this research area, also the paper is written as a research report (I would like to see that Result section is not so military structured). At some points some further clarifications should be made. To consider publication in this paper authors should address this concerns.

Reviewer 3 Report

In this study, the author investigated the effects of adopting hyperoxia and volatile anesthetics on various parameters including survival rate, inflammation and histopathological injury, in a rodent model of intestinal ischemia-reperfusion injury.

From a methodological point of view, in the first part of the study, the impact of hyperoxigenation was evaluated in intraperitoneal anesthesia with ketamine xylazine; in the second part of the study, I.P. anesthesia with ketamine xylazine was compared with isoflurane, both in conditions of hyperoxigenation. The author admits that this is a limitation of the study in the discussion; in fact, the fact that some experimental conditions are missing (the IRI isoflurane 20%, but also the sham I.P. 100%) does not allow drawing conclusions about the individual contribution of O2 and anesthetic agent.

The differences between ketamine/xylazine and isoflurane, as to their toxicological properties, should also be discussed, as well as a hypothesis about the toxicological mechanisms behind the differences observed.

Finally, the author should evidence the clinical implications, if there are any, of these findings.

Round 2

Reviewer 1 Report

The authors addressed all the issues proposed by the reviewers 

Reviewer 2 Report

Dear Authors, 

you gave fine explanations for my points.

Reviewer 3 Report

The authors addressed all the issues proposed by the reviewers in a sufficiently clear answer.